# Ag-MWCNT Composites for Improving the Electrical and Thermal Properties of Electronic Paste

**DOI:** 10.3390/polym16081173

**Published:** 2024-04-22

**Authors:** Yunkai Wang, Danlei Jing, Zikai Xiong, Yongqing Hu, Wei Li, Haotian Wu, Chuan Zuo

**Affiliations:** 1State Key Laboratory of Advanced Technologies for Comprehensive Utilization of Platinum Metals, Kunming Institute of Precious Metals, Kunming 650106, China; 2Sino-Platinum Metals Co., Ltd., Kunming 650106, China; wuht@ipm.com.cn; 3Sino-Platinum Electronic Materials (Yunnan) Co., Ltd., Kunming 650503, China

**Keywords:** electronic paste, carbon nanotube, electrical property, thermal property

## Abstract

With the development of microelectronics products with high density and high power, it is urgent to improve the electrical and thermal conductivity of electronic paste to achieve the new requirements of packaging materials. In this work, a new synthesis method of Ag-MWCNTs was designed: Firstly, carboxylated MWCNTs and stannous chloride were used as raw materials to prepare high-loading-rate Sn-MWCNT composite material to ensure the high loading rate of metal on the MWCNT surface. Then, Ag-MWCNT composite material was prepared by the chemical displacement method to solve the problem of the low loading rate of silver nanoparticles on the MWCNT surface. On the basis of this innovation, we analyzed and compared the electrical, thermal, and mechanical properties of Ag-MWCNT composite electronic paste. Compared with the electronic paste without adding Ag-MWCNTs, the resistivity was reduced by 77%, the thermal conductivity was increased by 66%, and the shear strength was increased by 15%. Therefore, the addition of Ag-MWCNTs effectively improves the electrical, thermal, and mechanical properties of the paste, making it a promising and competitive choice for new packaging materials in the future.

## 1. Introduction

Electronic adhesives, due to their excellent thermal and electrical properties, have been broadly used in the fields of microelectronic components and integrated circuits [1,2,3,4]. With the development of microelectronic products in the high-density and high-speed direction, most of the functions of an electronic system tend to be integrated into only one chip; therefore, the requirements for the electronic conductivity and thermal conductivity of the electronic pastes are gradually increasing.

Carbon nanotubes (CNTs) are considered excellent fillers for improving the electrical properties of electronic paste due to their outstanding electrical, thermal, and mechanical properties [5,6,7,8]. On the one hand, the excellent electronic and thermal conductivity of CNTs can efficiently improve the electronic and thermal conductivity of electronic paste. On the other hand, CNTs play a significant role in regulating the mechanical properties of the resin matrix in the paste [9,10,11,12]. Demski et al. [13] enhanced the impact resistance of acrylic resin by incorporating surface-functionalized multi-walled carbon nanotubes (MWCNTs). Anirudh et al. [14] investigated the influence of functionalized MWCNTs on the mechanical properties of an epoxy resin matrix. The results indicated that carboxylated MWCNTs improved the flexural strength and shear strength of the epoxy resin matrix, while amino-functionalized MWCNTs improved its tensile strength.

The depositing of metal nanometerials on the surface of carbon nanotubes has a wide range of applications in the field of electronic materials [15,16,17,18]. The connection between metal nanomaterials and carbon nanotubes is mainly achieved by electrostatic deposition and physical adsorption. Depositing nanometals on the surface of carbon nanotubes can effectively reduce the tendency of carbon nanotubes to aggregate [19]. There are some reports indicating that depositing nanosilver on the surface of carbon nanotubes can enhance the electronic conductivity and thermal conductivity of silver paste. Li et al. [20] synthesized Ag-MWCNT composite materials through chemical methods, which can improve the thermal conductivity of silver paste from 0.73 W/m·K to 0.96 W/m·K. Although the prepared composite material Ag-MWCNTs can improve the thermal conductivity of paste, the electrical resistivity of the paste has not been tested; moreover, the improvement in thermal conductivity is not significant, which cannot meet the requirements in microelectronic products. Suh et al. [21] prepared functionalized silver nanoparticles with phenyl groups, and silver nanoparticles were adsorbed onto carbon nanotubes through π-π interactions. The prepared Ag-MWCNTs can increase the electronic conductivity of silver paste from 1.12 × 10^4^ S·m^−1^ to 1.79 × 10^6^ S·m^−1^; meanwhile, the thermal conductivity can be increased from 1.64 W/m·K to 160 W/m·K. Although thermal conductivity and conductivity have been significantly improved, shear strength, as an important index of the safety in paste, has not been well measured. The silver content in the paste reached 80%, which will inevitably affect the shear strength of the paste.

The loading rate of silver nanoparticles on the surface of carbon nanotubes has a significant impact on the electrical and thermal properties of conductive adhesives. Therefore, improving the loading rate of silver nanoparticles on the surface of carbon nanotubes is the focus of this research direction. In this study, we designed a novel method for synthesizing Ag-MWCNTs, which utilizes the relationship between the strength of ion potential and the strength of complexation and solves the problem of the low loading rate of silver nanoparticles on the surface of carbon nanotubes. Carboxylated MWCNTs and stannous chloride were used as raw materials to prepare a high loading rate of Sn-MWCNT composite material to ensure the high loading rate of metal on the MWCNT surface. Then, through the chemical displacement of silver nitrate and tin, Ag-MWCNT composite material was finally obtained. This method has never been reported before. On the basis of this innovation, we added Ag-MWCNTs to the silver paste and systematically analyzed the influence of Ag-MWCNT composite material on the electrical, thermal, and mechanical properties of the silver paste. It was beneficial for the development and design of new packaging materials.

## 2. Materials and Methods

### 2.1. Materials

Carboxylated MWCNTs (purity > 95%, outer diameter 8–15 nm, length 50 µm) and stannous chloride were produced by Aladdin Biochemical Technologies (Shanghai, China), TT386 epoxy resin was produced by Tiantai High-tech (Guangzhou) Co., Ltd. (Guangzhou, China), SDS (sodium dodecyl sulfate) and SDBS (sodium dodecyl benzene sulfonate) (95%, mixture) were obtained from Sinopharm Chemical Reagent Co., Ltd. (Beijing, China), and flake silver powder and silver nitrate were produced by Kunming Institute of Precious Metals (Kunming, China). Polyether amine (D-230) curing agent was produced by Shandong Maifei Chemical Co., Ltd. (Jinan, China). Sodium borohydride was provided by Chengdu Kelong Chemical Reagent Factory (Chengdu, China).

### 2.2. Preparation of Ag-MWCNT Composites

A total of 25 mg of carboxylated MWCNTs, 0.5 g of stannous chloride, 50 mg of SDS, and 10 mL ethanol were added into beaker, and the mixture was stirred mechanically at a speed of 350 rpm for 4 h. Subsequently, 50 mL ethanol-saturated solution of sodium borohydride was gradually added to the mixture with continuous stirring for 30 min, and the solution was allowed to stand undisturbed for 6 h. After that, the supernatant in the beaker was decanted and subjected to three washes with deionized water before obtaining Sn-MWCNT nanofluids through centrifugation. A total of 10 mL deionized water was added to the obtained Sn-MWCNT nanofluid with stirring for 2 h. Subsequently, 50 mL silver nitrate solution (0.1 mol/L) was added to the solution and stirred continuously for additional 30 min. The obtained solution was allowed to stand for 6 h, followed by decanting the upper layer of liquid and subsequently washing the beaker with ethanol three times. The ethanol washing procedure effectively eliminated stannic acid and other hydrolysis by-products of tin ions. Finally, the Ag-MWCNT composite material was obtained after centrifuging and drying the raw product in the beaker.

### 2.3. Preparation of Modified Electronic Paste

The paste contained silver powder (7 g), TT386 epoxy resin (1.875 g), curing agent (1.125 g), and Ag-MWCNTs (0.1 g). The detailed production process is outlined as follows: Initially, the paste was prepared by premixing silver flakes, TT-386 epoxy resin, D-230 curing agent, and Ag-MWCNTs. Subsequently, the mixture was dispersed using a three-roll mill and defoamed under vacuum conditions. The resulting defoamed paste was printed onto a PET membrane and subjected to curing in an oven at 180 °C for 30 min followed by removal and cooling to room temperature.

### 2.4. Characterization

The surface morphology of Ag-MWCNTs was observed using a Transmission Electron Microscope (TEM). X-ray diffraction (XRD) was employed to detect the generation of silver on carbon nanotubes. Thermogravimetry (TG) was utilized to analyze the decomposition temperature of carbon nanotubes and epoxy resin. Differential Scanning Calorimeter (DSC) was applied to analyze the melting temperature of silver nanotubes on carbon nanotubes. The thermal conductivity of modified paste was measured using a thermal conductivity meter. Shear strength of paste was determined using an electronic universal testing machine. A multimeter was utilized for measuring the resistance of paste.

## 3. Results and Discussion

### 3.1. Morphological Analysis of Sn-MWCNT Composites

The impact of MWCNTs with distinct functional groups on the morphology of Ag-MWCNT composites was assessed by comparing the microstructures of Ag-MWCNT composites derived from untreated MWCNTs, carboxylated MWCNTs, and hydroxylated MWCNTs. SDS served as a dispersant while sodium borohydride ethanol solution acted as a reducing agent. As shown in Figure 1a, Sn-MWCNT composites obtained from untreated MWCNTs exhibited lower surface loading of tin nanoparticles, presenting a smooth state primarily, and numerous carbon nanotubes were aggregated together. Due to the inherent stability of the surface of non-functionalized carbon nanotubes, it is difficult for metal ions to attach to the surface without active sites, which affects the loading efficiency of silver nanoparticles. The successful loading of tin nanoparticles onto the surface of functionalized MWCNTs is evident from Figure 1b,c. However, the carboxylated MWCNTs exhibited a higher loading rate of tin nanoparticles compared to hydroxylated MWCNTs, with a reduced amount of free tin. The functionalization process rendered the surface of MWCNTs more reactive, facilitating the effective attachment of other atoms or groups. Although both carboxyl and hydroxyl groups on the surface of carbon nanotubes could form stable complexes with metal ions, carboxyl groups possess an additional adjacent double-bond oxygen compared to hydroxyl groups. Therefore, one oxygen atom is utilized when forming a complex between hydroxyl groups and metal ions, while carboxyl groups can provide two oxygen atoms. Moreover, in general, complexes formed by carboxyl groups may be more stable, resulting in denser silver deposition on the MWCNT surface during subsequent reduction and directly affecting the loading rate of silver nanoparticles on carbon nanotubes.

The MWCNTs exhibited hydrophobic properties, and the presence of strong van der Waals forces promoted their aggregation into clusters or polymers within the system, significantly influencing the loading behavior of tin nanoparticles and silver nanoparticles. Therefore, for the preparation of Sn-MWCNT composites, dispersants should be incorporated into the system for MWCNT treatment, while the non-covalent surface modification of carbon nanotubes was essential to facilitate their disaggregation, ensuring the homogeneous dispersion of carbon nanotubes and preserving their original aspect ratio and electronic structure. SDS and SDBS were utilized as dispersants during the preparation of Sn-MWCNT composites.

As illustrated in Figure 2, the dispersion effect of Sn-MWCNTs prepared with SDS dispersant was significantly superior to that of the other two. SDS and SDBS were two commonly employed anionic surfactants, frequently utilized for the dispersion of carbon nanotubes. Anionic surfactants could facilitate the dispersal of carbon nanotubes by dissociating the negatively charged groups on their surface, thereby promoting effective disaggregation and uniform dispersion within the system. The SDBS dispersant exhibited superior performance compared to SDS in the synthesis of Ag-MWCNTs through the direct reduction of AgNO_3_. Specifically, the benzene ring structure in SDBS could form specific π-π stacking interactions with carbon nanotubes, resulting in non-covalent bond formation, which effectively disaggregated carbon nanotubes and prevented agglomeration in the solution. However, in the experiment, the addition of stannous chloride to MWCNTs was chosen for the synthesis of Ag-MWCNTs through reduction and substitution processes. Upon adding stannous chloride, a significant amount of chloride ions were present in the solution, resulting in the sulfonic acid in SDBS reacting with these chloride ions, finally leading to a considerable decrease in the dispersion ability of SDBS. Consequently, the agglomeration of carbon nanotubes within the system also impacted the loading rate of metal on the surface of MWCNTs.

### 3.2. Morphology Analysis and Characterization of Ag-MWCNT Composites

The quantity and morphology of silver nanoparticles on the surface of carbon nanotubes were influenced by the addition of dispersants, functional groups, and stannous chloride. The investigation was conducted to examine the impact of stannous chloride addition on the morphology of Ag-MWCNT composites. Four experiments were conducted for comparison, with varying amounts of stannous chloride added: 0.2 g, 0.5 g, 1 g, and 1.5 g. The inclusion of other substances remained unaltered, with the exception of the incorporation of stannous chloride.

As illustrated in Figure 3, when the addition amount of stannous chloride is 0.2 g, the silver content in the system is relatively low. Despite the small particle size of silver nanoparticles on carbon nanotube surfaces, the loading quantity of silver nanoparticles remained insufficient to effectively enhance the electrical and thermal conductivity of the paste in subsequent usage. On the other hand, with stannous chloride contents of 1 g and 1.5 g, a higher loading quantity of silver nanoparticles onto the carbon nanotubes’ surface was achieved, which resulted in larger silver nanoparticles along with increased free silver ions. Consequently, when the replacement reaction occurred, tin predominantly existed in stable +4 ionic form, and there was a rapid expansion in silver content leading to the significant agglomeration of silver nanoparticles within the system. Conversely, when stannous chloride content was too low, reduced tin concentration failed to support substantial loading of silver nanoparticles on the surface of carbon nanotubes. Especially after reducing agent addition, tin ion concentration rapidly decreased, making the reduction reaction more challenging. Based on these findings, it was concluded that Ag-MWCNTs prepared using an addition amount of stannous chloride of 0.5 g would be utilized for subsequent analysis.

The Energy Dispersive Spectrometer (EDS) was primarily utilized for the elemental analysis of minuscule regions within materials. In this study, energy spectrum analysis was performed on the silver nanoparticles, as depicted in Figure 4, notably, a trace amount of tin nanoparticles was detected within the silver nanoparticles loaded onto the surface of carbon nanotubes, which was attributed to the preferential reactivity of tin atoms on the surface during the replacement reaction. With the reaction progressing, silver atoms gradually loaded the tin surface, hindering further contact between internal tin atoms and the silver nitrate solution. Therefore, a small amount of tin atoms remained within the silver structure, accounting for approximately 8% of total metal atoms, the influence of which on overall performance was negligible.

The successful formation of silver and tin nanoparticles on the surface of carbon nanotubes was characterized using X-ray diffraction (XRD). XRD images of carboxylated MWCNTs, MWCNTs loaded with tin nanoparticles, and MWCNTs loaded with silver nanoparticles are depicted in Figure 5. The XRD results of carboxylated MWCNTs are depicted in curve (a), wherein the dominant peak corresponds to the crystal plane (002) of the carbon nanotubes. The XRD patterns reveal that the peaks observed at 2θ = 30.64°, 32.00°, 43.84°, and 44.06° in curve (b) correspond to the crystal planes of Sn indexed as (200), (101), (220), and (211), respectively, confirming the presence of Sn-MWCNTs. The peak of tin exhibits sharpness without any additional impurity peaks, indicating the excellent crystallinity of tin and the absence of the oxidation of tin nanoparticles on the surface of carbon nanotubes. The aforementioned favorable condition facilitates subsequent silver replacement reactions occurring on the carbon tube surface. The XRD patterns of Ag-MWCNTs are presented in curve (c). Upon comparing the XRD cards, it can be observed that the peaks at 2θ = 38.08°, 44.24°, 64.44°, and 77.38° correspond to the crystallographic planes (111), (200), (220), and (311), respectively. Notably, no distinct peak of tin is detected in the spectrum, indicating the successful replacement of tin and the absence of silver oxidation. The sharp peak indicates enhanced silver crystallinity, and the grain crystallinity was estimated by fitting the half-peak width. The results showed that the final crystallinity of silver nanotubes on carbon nanotube surfaces reached 72.31%. By comparing the XRD results of Ag-MWCNTs and Sn-MWCNTs, it was evident that the peak of silver significantly surpassed that of tin due to a valence state issue. Tin tended to replace more silver, resulting in a higher content of silver than tin and an elevated silver peak.

The decomposition temperature of carbon nanotubes was characterized by thermogravimetric (TG) analysis. TG analysis was performed separately on carboxylated MWCNTs, Sn-MWCNTs, and Ag-MWCNTs. As the TG results show in Figure 6a for carboxylated MWCNTs, the decomposition of carbon nanotubes initiated at approximately 330 °C, with an accelerated rate at 420 °C and reaching its maximum value at 567 °C. According to the TG result comparison presented in Figure 6b, it can be observed that the carbon nanotubes loaded with silver exhibited an onset of decomposition at a temperature of 256 °C, followed by a subsequent increase in decomposition rate, reaching its maximum at 445 °C. When the temperature reached 290 °C, the tin-loaded carbon nanotubes exhibited decomposition, with the decomposition rate peaking at 595 °C. By comparing the initial decomposition temperature, it can be inferred that metal loading on the surface of carbon nanotubes induced structural damage. Although metal loading reduced the decomposition temperature, Ag-MWCNTs exhibited a significantly higher decomposition temperature than the paste curing temperature and did not cause any damage to the carbon nanotubes during curing.

The DSC method was carried out to analyze the silver melting point on the surface of carbon nanotubes, which can be utilized for creating an electrical and thermal conductive pathway by combining carbon nanotubes with silver powder in paste. As shown in Figure 7, there is a significant endothermic peak at 36 °C, which may be related to the volatilization of the solvent. With increasing temperature, subsequent smaller sawtooth-shaped heat absorption peaks emerged, corresponding to varying particle sizes of nanosilver within the system.

### 3.3. Analysis of Modified Paste

TG analysis was also performed on the pastes containing Ag-MWCNTs, and the results are presented in Figure 8. As the temperature increased, a slight decrease in weight was observed initially, which can be attributed to water evaporation. Upon reaching the curing temperature of 180 °C, no further change in weight occurred, indicating that the structure of paste remained intact within this temperature range. However, as the temperature reached 210 °C, a continuous decrease in weight was observed, suggesting decomposition of a portion of the epoxy resin. When the temperature reached 355 °C, the decomposition rate of the paste peaked. As the temperature continued to rise, the epoxy resin underwent substantial decomposition at around 400 °C, and its decomposition rate gradually approached zero. Ultimately, at 500 °C, a residual mass of 73.78% remained.

The resistivity tests were conducted on the initial paste, as well as the modified pastes containing carboxylated MWCNTs and Ag-MWCNTs, with both additives added at a concentration of 1%. As depicted in Figure 9, the inclusion of carboxylated MWCNTs had a negligible impact on the overall resistivity of the paste, whereas the incorporation of Ag-MWCNTs significantly reduced its resistivity. This phenomenon can be attributed to the strong van der Waals forces between carbon nanotubes in paste and the high aspect ratio of carbon nanotubes. Therefore, upon addition to the system, carbon nanotubes tended to entangle and aggregate, creating substantial barriers for effective contact between flaky silver powder particles within the paste. In contrast to carboxylated MWCNTs, the presence of silver-nanoparticle-functionalized Ag-MWCNTs resulted in increased steric hindrance, thereby reducing the possibility of carbon nanotube entanglement and reaggregation. Additionally, the sintering of silver nanoparticles on the surface of carbon nanotubes with sheet-like silver powder in the paste facilitated the formation of more conductive pathways without requiring high temperatures, effectively harnessing the inherent high conductivity of carbon nanotubes.

Furthermore, the impact of incorporating Ag-MWCNTs on the resistivity of the modified paste was investigated. The alterations in resistivity for the initial paste containing 0.5%, 1%, 1.5%, and 2% Ag-MWCNT composites are depicted in Figure 10. The results indicate that the addition amount of Ag-MWCNT composite material did not exhibit a linear relationship with the resistivity of the modified paste. Based on a series of experiments, it was found that when 1% of Ag-MWCNT composite material was added, the conductivity of the paste reached its maximum improvement. For further investigations on the thermal conductivity and shear strength of the modified paste, a 1% addition of Ag-MWCNTs in the modified paste was utilized.

The comparative analysis of the thermal conductivity between the original paste, the modified paste incorporating carboxylated MWCNTs, and Ag-MWCNTs was conducted. As shown in Figure 11, the thermal conductivity values for the original paste, MWCNT-enhanced paste, and Ag-MWCNT-enhanced paste were measured as 4.96 W/mK, 6.07 W/mK, and 8.25 W/mK, respectively. The observed enhancement in thermal conductivity for the Ag-MWCNT-enhanced paste can be primarily attributed to the presence of silver nanoparticles on the surface of carbon nanotubes. The thermal conductivity of conventional paste primarily relied on the heat transfer occurring between silver powders. However, when carbon nanotubes were introduced into the paste, their exceptional thermal conductivity was not fully utilized due to the inadequate contact with the sheet-like silver powders in the paste. By incorporating silver nanoparticles onto the surface of carbon tubes, a solid phonon transmission junction and an efficient phonon transmission channel can be established during the curing process, facilitating the contact between carbon tubes and silver powder in the paste. This approach reduces the carrier transmission barrier between fillers and enhances carrier interface transmission efficiency. In contrast, carbon nanotubes can serve as an effective thermal conductive bridge between silver powders, facilitating the formation of multiple pathways for efficient heat transfer. With heat absorption, silver nanoparticles located on the surface of carbon nanotubes exhibited rapid migration to the carbon nanotube structure, which was attributed to the exceptional phonon average free path exhibited by carbon nanotubes, enabling swift and widespread dispersion of thermal energy.

Shear strength tests on the initial paste and the modified paste with Ag-MWCNTs were also performed. As depicted in Figure 12, the addition of Ag-MWCNT composite material led to a significant enhancement in the shear strength of the paste, specifically increasing from 7.82 MPa to 9.01 MPa. The distinct improvement can be attributed to two primary influencing factors: firstly, through sintering, silver functionalized nanotubes adhered firmly onto the surface of carbon nanotubes and flaky silver powder in the paste, resulting in a strong bond between carbon nanotubes and the paste; secondly, owing to their inherent high strength properties, carbon nanotubes themselves contributed to enhancing the shear strength of the modified paste. On the other hand, the incorporation of carbon nanotubes can enhance epoxy resin reactivity, elevate epoxy resin crosslinking degree, and hence improve shear strength.

## 4. Conclusions

The Ag-MWCNTs were synthesized through a chemical displacement method, and TEM images revealed the homogeneous distribution of silver nanoparticles on the surface of MWCNTs. Upon incorporating 1%wt Ag-MWCNTs into the paste, a significant reduction in resistivity from 2.49 × 10^−6^ Ω·m to 0.58 × 10^−6^ Ω·m was observed, accompanied by a distinct enhancement in thermal conductivity from 4.96 W/mK to 8.25 W/mK and an increase in shear strength by 1.19 MPa. The enhanced electrical and thermal conductivity of the cured paste can be attributed to the surface nanosilver of Ag-MWCNTs. During the curing process, sintering occurred between the Ag-MWCNT surface nanosilver and silver powder in the paste, thereby establishing an efficient conductive pathway for electricity and heat transfer. This synergistic effect fully exploits the exceptional mechanical properties of carbon nanotubes. In conclusion, the synthesized Ag-MWCNTs can significantly enhance the overall performance of the paste including thermal conductive, electrical conductive, and mechanical properties, thereby meeting the performance requirements of current microelectronic products.

## Figures and Tables

**Figure 1 polymers-16-01173-f001:**
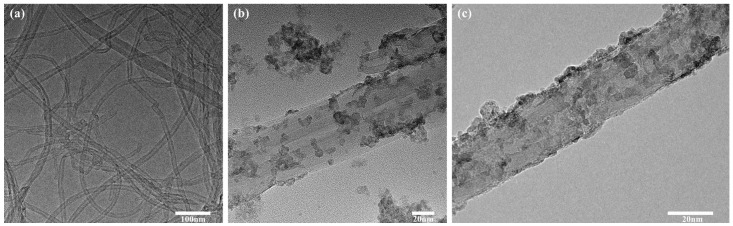
Sn-MWCNTs prepared from different MWCNTs: (**a**) Sn-MWCNTs prepared from unfunctionalized MWCNTs; (**b**) Sn-MWCNTs prepared from hydroxylated MWCNTs; (**c**) Sn-MWCNTs prepared from carboxylated MWCNTs.

**Figure 2 polymers-16-01173-f002:**
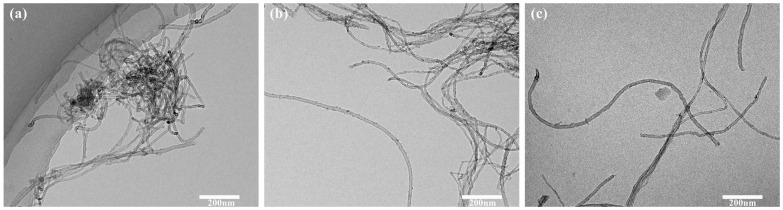
Sn-MWCNTs prepared with different dispersants. (**a**) The application of dispersants was not employed. (**b**) The dispersant used is SDBS. (**c**) The dispersant used is SDS.

**Figure 3 polymers-16-01173-f003:**
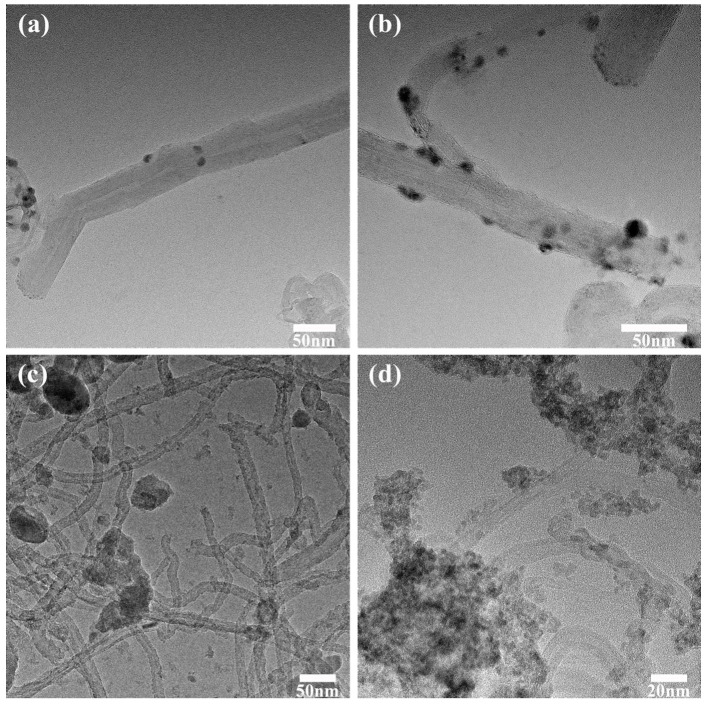
The influence of stannous chloride dosage on the morphology of Ag-MWCNTs. (**a**) The addition amount of stannous chloride is 0.2 g. (**b**) The addition amount of stannous chloride is 0.5 g. (**c**) The addition amount of stannous chloride is 1.0 g. (**d**) The addition amount of stannous chloride is 1.5 g.

**Figure 4 polymers-16-01173-f004:**
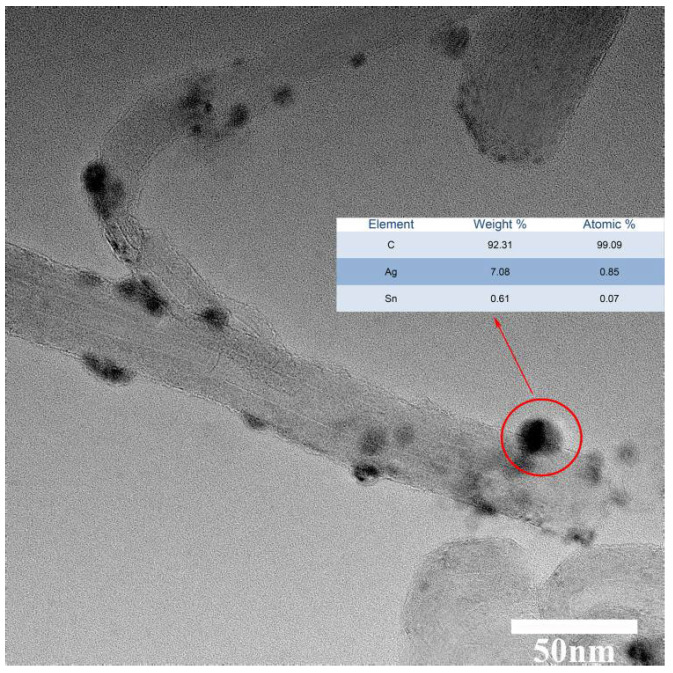
Metal content on the surface of carbon nanotubes.

**Figure 5 polymers-16-01173-f005:**
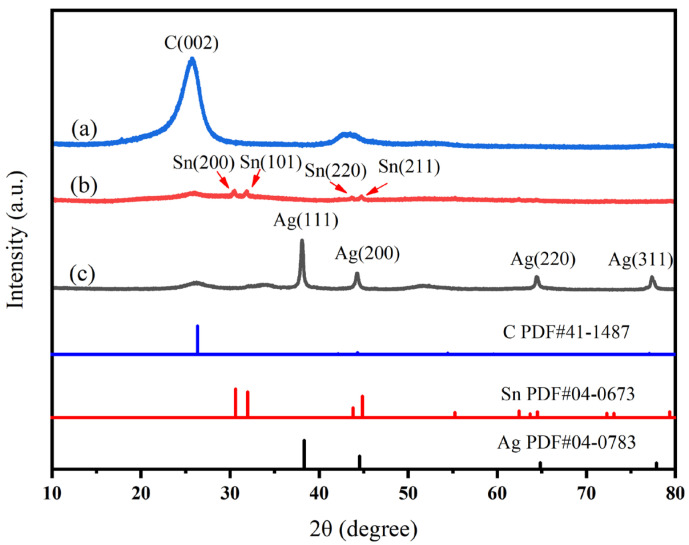
The XRD curves of (**a**) carboxylated MWCNTs, (**b**) Sn-MWCNTs, and (**c**) Ag-MWCNTs.

**Figure 6 polymers-16-01173-f006:**
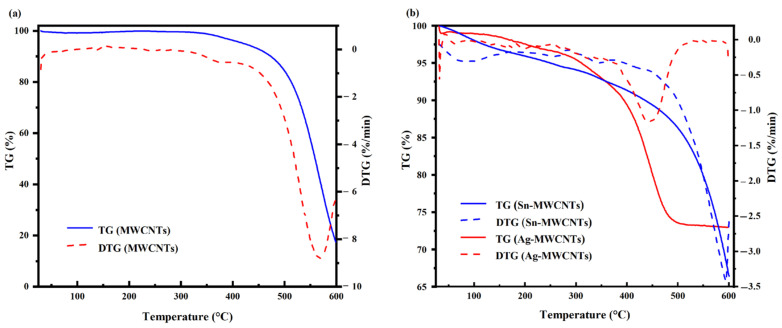
TG curves of (**a**) carboxylated MWCNT, (**b**) Sn-MWCNT, and Ag-MWCNT composites.

**Figure 7 polymers-16-01173-f007:**
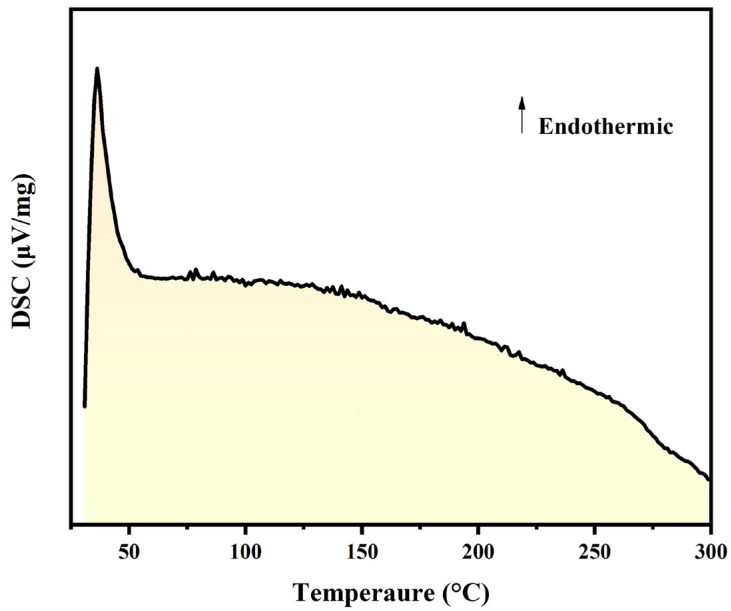
DSC curve of Ag-MWCNTs.

**Figure 8 polymers-16-01173-f008:**
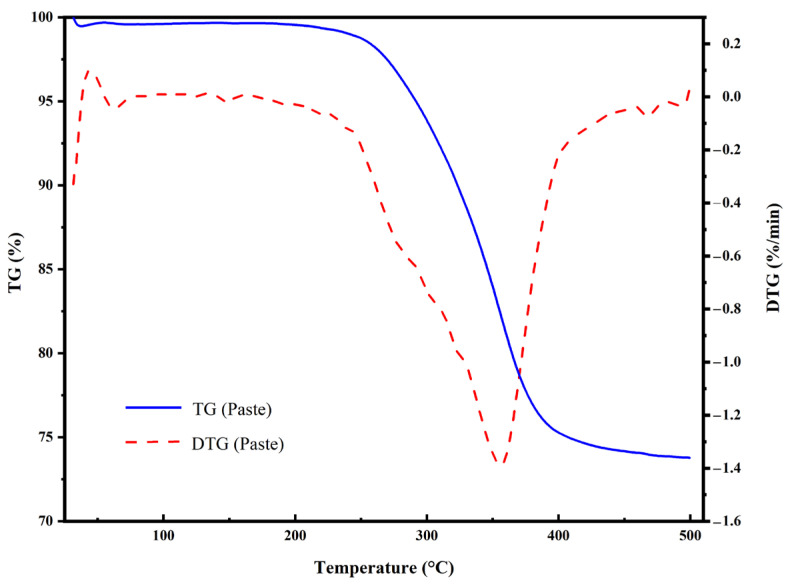
Results of thermogravimetric analysis for the modified pastes obtained.

**Figure 9 polymers-16-01173-f009:**
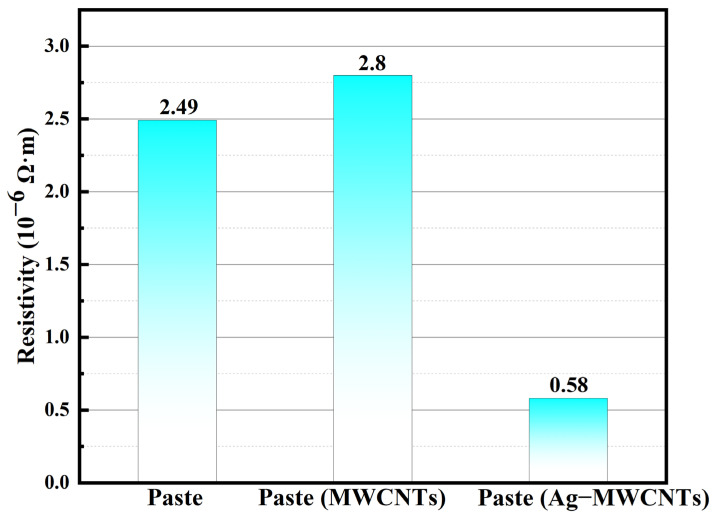
Resistivity of three kinds of pastes.

**Figure 10 polymers-16-01173-f010:**
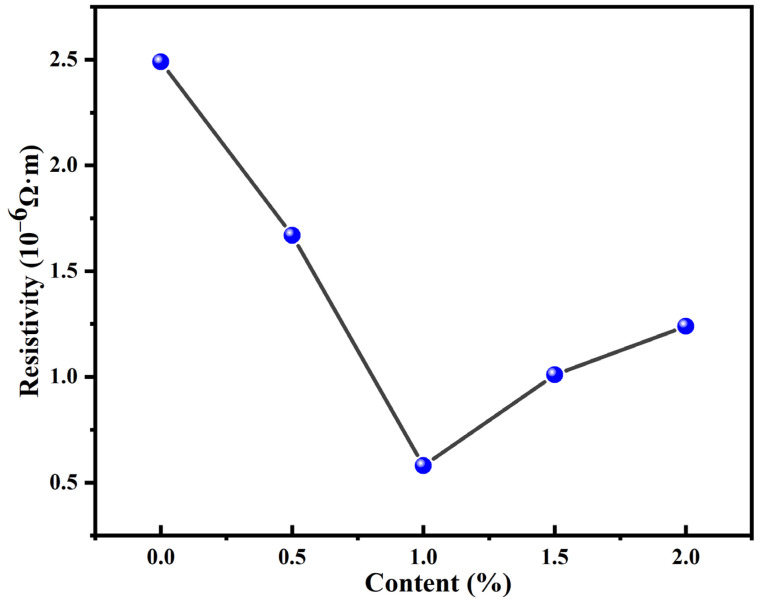
Effects of paste resistance through the addition of Ag-MWCNTs.

**Figure 11 polymers-16-01173-f011:**
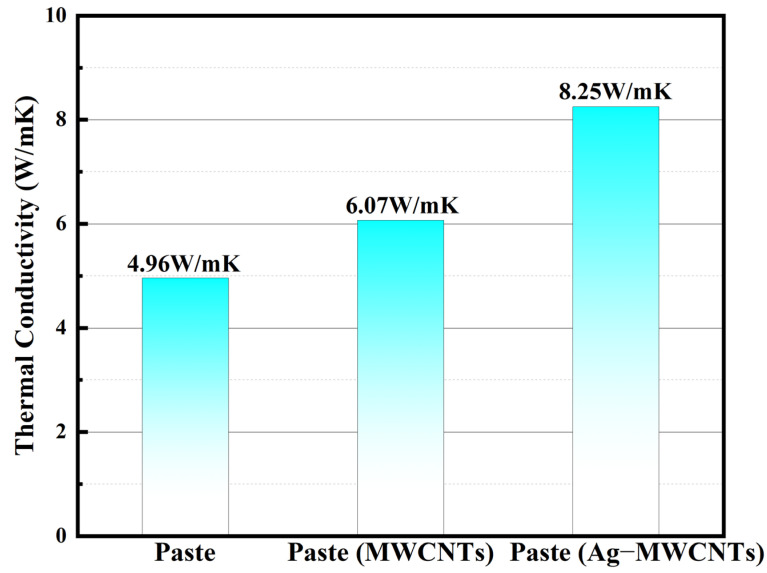
Thermal conductivity of three kinds of paste.

**Figure 12 polymers-16-01173-f012:**
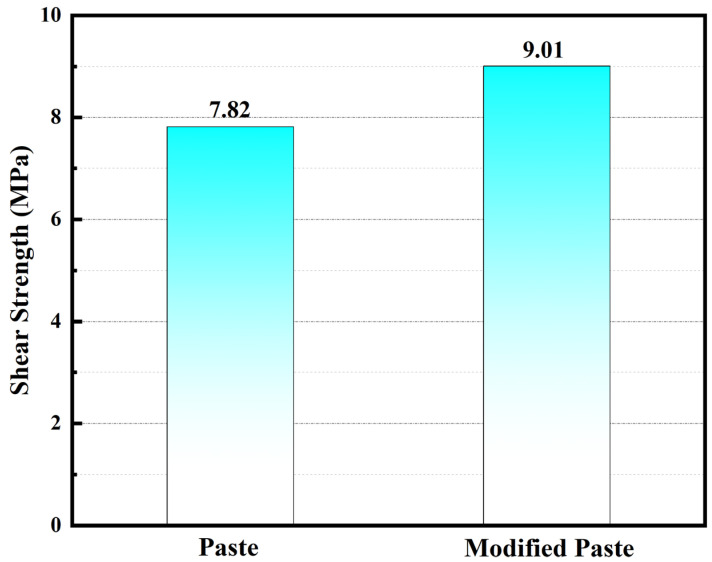
Shear strength of two types of pastes.

## Data Availability

All data are available in the main text.

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
