# Peer review of "Ag-MWCNT Composites for Improving the Electrical and Thermal Properties of Electronic Paste"

_polymers, 2024, doi:10.3390/polym16081173_

Round 1

Reviewer 1 Report

Comments and Suggestions for Authors

1. Please revise the title, including the entire manuscript, particularly "Ag@MWCNTs," to "Ag-MWCNTs." 

2. The abstract's last statement concerns overall performance; what does this mean?  The author should write the properties or specific performance.

3. In the introduction section, the authors mentioned the large citation group needs to keep a low range of citation groups like [5-8] instead of [5-13].

4. Please re-write the statement, "In this study, a layer of nano tin was first loaded onto the surface of carbon nanotubes, and then Ag@MWCNTs were prepared by replacement tin with silver."

5. Throughout the manuscript, please make the changes in the unit used, such as 0.5g, hours should be Hr, and in a few places, they mention "paste contained 7 g of silver powder, 1.875 g"

6. Please re-write the statement, "This is attributed to the inherent stability of unloaded carbon nano-tube surfaces, which hinders the absorption of metal ions and consequently influences the subsequent loading efficiency of silver nanoparticles through displacement reactions."

7. Please check this "AgNO3" and correct it. 

8. Please explain if any additional impurity peaks are shown in XRD images.

9. Figure 5 (c) shows the small peaks reflecting; could you mark it or do the rebuttal? 

10. Is there any reason it has a 300-degree Celsius temperature range in Figure 7?

Reviewer 2 Report

Comments and Suggestions for Authors

The manuscript is generally intriguing, but its presentation is unclear in several instances. Here are some suggestions to improve:

1.      Abstract: The abstract should only cover the current topic of the paper. To effectively engage readers, authors should present significant results in their abstracts.

2.      The author adequately emphasizes the novelty of the current work at the end of the introduction.

3.      Figure 5 should include the y-axis unit in the XRD pattern. Please compare it to the standard JCPDS file, and include the results in Figure 5.

4.      Why did the author choose chemical displacement over other conventional methods? Is there a specific mechanism underlying this choice?

5.      The primary findings of Sn@MWCNTs should be included in the abstract.

6.      Adding the DSC curve of Sn@MWCNTs would enhance the revised manuscript.

7.      If it’s possible to investigate the electronic structure of the prepared samples through XPS analysis which can improve the quality of the manuscript.

Comments on the Quality of English Language

1.      Typographical errors are present throughout the manuscript. Authors are required to pay keen attention to this
